# High-Performance HVOF-Sprayed Fe-Based Amorphous Coating on LA141 Magnesium Alloy via Optimizing Oxygen Flow and Kerosene Flow

**DOI:** 10.3390/ma14174786

**Published:** 2021-08-24

**Authors:** Yijiao Sun, Weichao Wang, Hongxiang Li, Lu Xie, Yongbing Li, Shanlin Wang, Wenrui Wang, Jiaming Zhang, Jishan Zhang

**Affiliations:** 1State Key Laboratory for Advanced Metals and Materials, University of Science and Technology Beijing, Beijing 100083, China; sunyijiao0920@126.com (Y.S.); w18801090723@163.com (W.W.); zhangjs@skl.ustb.edu.cn (J.Z.); 2School of Mechanical Engineering, University of Science and Technology Beijing, Beijing 100083, China; xielu@ustb.edu.cn (L.X.); zhangjm@ustb.edu.cn (J.Z.); 3State Key Laboratory of Advanced Forming Technology and Equipment, Beijing National Innovation Institute Lightweight Ltd., Beijing 100083, China; lybustb@163.com; 4School of Aeronautical Manufacturing Engineering, Nanchang Hangkong University, Nanchang 330063, China; slwang70518@nchu.edu.cn

**Keywords:** LA141 magnesium alloy, Fe-based amorphous coatings, high-velocity oxygen-fuel (HVOF) spraying, corrosion and wear resistance, bonding strength

## Abstract

To solve the problem of poor corrosion and wear resistance of Mg-Li alloys, Fe-based amorphous coatings were prepared by high velocity oxygen-fuel spraying technology (HVOF) on the LA141 magnesium alloy substrate with a Ni60 intermediate layer. The microstructure and performance of Fe-based amorphous coatings with different oxygen flow and kerosene flow were characterized and analyzed. The results demonstrate that there is an optimal oxygen/kerosene ratio where the porosity of Fe-based amorphous coating is the lowest. Moreover, the amorphous content increases with the decrease in the oxygen/kerosene ratio. In particular, when the oxygen flow is 53.8 m^3^/h and the kerosene flow is 26.5 L/h, the Fe-based amorphous coating possesses the lowest porosity (0.87%), the highest hardness (801 HV_0_._1_), the highest bonding strength (56.9 MPa), and an excellent corrosion and wear resistance. Additionally, it can be seen that the Fe-based amorphous coating is composed of amorphous splats and amorphous oxides, but the Ni60 intermediate layer exhibits an amorphous and crystalline multi-phase structure. The high bonding strength of the coating is attributed to the low porosity of Fe-based amorphous coating and the localized metallurgical bonding between different layers. Finally, the mechanisms on corrosion and wear of Fe-based amorphous coatings are also discussed.

## 1. Introduction

As a class of promising ultra-lightweight materials, Mg-Li alloys have attracted much attention because they can significantly reduce the weight of an aircraft by replacing Al or Ti alloy components. However, because of their high electrochemical activity and low hardness, the Mg-Li alloys often suffer from corrosion in the service environment and abrasion from working conditions at the same time, especially in a humid and hot oceanic climate [1,2]. Surface treatment is an effective means to improve the corrosion and wear resistance of Mg-Li alloys, such as micro-arc oxidation (MAO), chemical conversion, painting, electroplating and electroless plating. Nevertheless, these methods either cannot provide satisfactory corrosion and wear protection for Mg-Li alloys simultaneously, or require complex pre-treatment and post-treatment processes [3,4,5,6]. Therefore, it is urgent to develop a coating with a simple preparation process and excellent corrosion and wear resistance on the surface of Mg-Li alloys.

On account of the short-range ordered and long-range disordered structure, the absence of crystalline defects, the homogeneity of the chemical composition, and the presence of passivity promoters and dissolution blockers, Fe-based amorphous alloys exhibit high strength, high hardness, high elastic strain, and preeminent corrosion and wear resistance [7,8,9]. However, it is difficult to produce Fe-based amorphous alloys with large sizes because the formation of amorphous alloys needs a great cooling rate and strict preparation conditions. Fe-based amorphous metallic coatings (denoted as Fe-based AMCs hereafter) can not only solve the problem of limited glass formation ability, but also give full play to the excellent corrosion and wear resistance of Fe-based amorphous alloys [10,11,12]. There have been many investigations on the preparation and performance of Fe-based AMCs on steels. For instance, Tian et al. reported that the corrosion current density and potential of Fe-based AMC on mild steel is equivalent to that of the hard chromium coating, but the passivation capability of the Fe-based AMC is significantly higher than that of the hard chromium coating [13]. Zhang et al. found that the Fe-based AMC on mild steel shows a better corrosion resistance than 304 SS and X80 steel in sulfate-reducing bacteria (SRB) corrosion due to the toxicity of Mo, W, and Ni elements and the compact passive film [14]. Zheng et al. reported that the Fe-based AMC on 304 SS exhibits much higher resistance to erosion-corrosion than the substrate in sand-containing NaCl solution because the Fe-based AMC possesses higher hardness and better re-passivation ability [15]. In addition, Fe-based AMC has also been used recently to provide corrosion and wear protection for commercial magnesium alloys. Guo et al. prepared the Fe-based AMC on AZ61 magnesium alloy with a NiCrAl intermediate layer. The result shows that the corrosion resistance of Fe-based AMC is superior to most coatings on magnesium alloys and the coating exhibits a relatively high bonding strength (40 MPa) [16]. Yuan et al. fabricated the Fe-based AMC on AZ31B magnesium alloy directly. Compared with the AZ31B magnesium alloy substrate, the Fe-based AMC not only shows a lower friction coefficient, but also exhibits an extremely low wear rate [17]. To sum up, Fe-based AMC possesses excellent corrosion resistance and wear resistance, but there are almost no reports about the Fe-based AMC on Mg-Li alloys, although it can also provide a favorable corrosion and wear protection for Mg-Li alloys.

In the present work, HVOF spraying technology was utilized to prepare Fe-based AMCs with a Ni60 intermediate layer on the LA141 magnesium alloy substrate. In order to achieve the high-performance Fe-based AMC, the oxygen flow and kerosene flow are optimized and their impact on the porosity, amorphous content, hardness, bonding strength, and corrosion and wear behaviors of Fe-based AMCs were investigated in detail. Finally, the mechanisms on corrosion and wear of Fe-based AMCs were also discussed.

## 2. Materials and Methods

Fe-based amorphous powders (Fe_48_._8_Cr_23_._4_Mo_19_._8_Si_5_C_2_._1_B_0_._9_, wt. %) and commercial Ni60 powders (15–45 μm) were deposited on LA141 magnesium alloy substrate with the size of 100 mm × 100 mm × 10 mm by HVOF spraying to fabricate Fe-based AMC with a Ni60 intermediate layer. All the LA141 magnesium alloy plates were polished by sandpaper, cleaned by ethanol and sandblasted by quartz sand before spraying. The Ni60 intermediate layer with a thickness of about 150 μm was prepared on the LA141 magnesium alloy plates, and then the Fe-based AMC was fabricated on the Ni60 intermediate layer. The spraying parameters of the Ni60 intermediate layer were as follows: the oxygen flow was 51.0 m^3^/h, the kerosene flow was 22.7 L/h, the spray distance was 350 mm, the powder feed rate was about 80 g/min, and the scanning velocity was 300 mm/s. The oxygen flow and kerosene flow were optimized to obtain a high performance of Fe-based AMC, and the detailed spraying parameters of Fe-based AMCs are shown in Table 1.

X-ray diffraction analysis of Fe-based amorphous powders was carried out using monochromatic Cu Kα radiation on a Bruker D8 advance diffractometer (XRD, Bruker Corporation, Billerica, MA, USA). The microstructure and chemical compositions of Fe-based AMCs, Ni60 intermediate layers, LA141 magnesium alloy, and Fe-based amorphous powders were observed and analyzed by a scanning electron microscope (SEM, Jeol Ltd., Tokyo, Japan) equipped with an energy dispersive spectrometry detector (EDS). The enthalpy changes of Fe-based amorphous powders and Fe-based AMCs were measured by a differential scanning calorimetric (DSC, TA Instruments, New Castle, DE, USA) with a heating rate of 10 ℃/min. The amorphous content of Fe-based AMC was the ratio of the exothermic enthalpy of Fe-based AMC to that of amorphous powders. The amorphous content of Fe-based AMC ($P_{coating}$) can be calculated and the calculation formula is
(1)$$P_{coating}=\left(∆H_{coating}/∆H_{powder}\right)\times 100\%$$
where $∆H_{coating}$ and $∆H_{powder}$ are the total exothermic enthalpy of Fe-based AMC and Fe-based amorphous powders after being fully crystallized, respectively; ten cross-sectional optical photographs (500×) of Fe-based AMCs were used to calculate the porosity by Image-Pro software. The optical photographs were obtained using an optical microscope (Carl Zeiss AG, Oberkochen, Germany) and the porosity of Fe-based AMC was the ratio of the pore area to the total region. The high-resolution transmission electron microscopy (HRTEM, FEI Company, Hillsboro, OR, USA) was utilized to observe the microstructure of Fe-based AMC and the Ni60 intermediate layer.

A Vickers Hardness Tester (Yashitejiu Instruments Corporation, Hangzhou, China) was used to measure the microhardness of Fe-based AMCs, and the applied load is 0.98 N. According to ASTM C633-13(2017) standard [18], the bonding strength between Fe-based AMCs and the LA141 magnesium alloy substrate were measured by an electronic universal testing machine (Huage Instruments Corporation, Jinan, China). The corrosion behaviors of Fe-based AMCs and LA141 magnesium alloy substrate in 3.5 wt. % NaCl solution were evaluated by an electrochemical workstation (Corrtest Instruments Corporation, Wuhan, China). The potentiodynamic measurements were performed with a scanning rate of 0.5 mV/s and the potentiostatic measurements were carried out at the polarization potentials of 0.5 V_SCE_ for 5000 s. Electrochemical impedance spectroscopy (EIS) of Fe-based AMC and LA141 magnesium alloy substrate were tested at open circuit potential with a sinusoidal amplitude of 10 mV, and the EIS data were fitted by the Zview Version: 2.80 software. The sliding wear behaviors of Fe-based AMCs and LA141 magnesium alloy substrate were measured by a ball-on-disk tribometer (Bruker Corporation, Billerica, MA, USA) without lubrication. The friction pairs were Si_3_N_4_ balls with a diameter of 4 mm. The rotating speed, rotating radius, applied load, sliding time, and sliding distance were 400 r/min, 3 mm, 5 N, 20 min, and 150.8 m, respectively. The volume losses of Fe-based AMCs and LA141 magnesium alloy were measured by a white-light interferometer (Bruker Corporation, Billerica, MA, USA). The wear rates were calculated using the equation of
(2)$$Q=\frac{V}{NS}$$
where *V* was the wear loss volume (mm^3^), *N* was the applied load (N), *S* was the total sliding distance (m) and *Q* was the wear rate (mm^3^/(N m)), respectively. Bonding strength tests, wear tests and electrochemical measurements were repeated three times to ensure the reliability of data.

## 3. Results

Figure 1 displays the surface morphology, size distribution, and XRD pattern of Fe-based amorphous powders produced by gas atomization. The size of Fe-based amorphous powders ranges from 5 μm to 50 μm, and the most powders have a smooth surface and a spherical or near-spherical shape. Due to the limitation of XRD, the typical broad diffraction hump in the XRD pattern can prove that the Fe-based amorphous powders are mostly amorphous.

The cross-sectional morphology and surface morphology of the Fe-based AMCs with different oxygen flow and kerosene flow are shown in Figure 2. The coatings on the LA141 magnesium alloy substrate can be divided into two layers, i.e., the outermost layer is Fe-based AMC, and the intermediate layer is the Ni60 coating. There are some pores in both Fe-based AMC and the Ni60 intermediate layer. Whatever the spraying parameters are, there are no cracks and pores at the Fe-based AMC/Ni60 layer interface and the Ni60 layer/LA141 alloy substrate interface. Additionally, the roughness of the Ni60 layer/LA141 alloy substrate interface is obviously higher than that of the Fe-based AMC/Ni60 layer interface due to the sand blasting of the LA141 alloy substrate surface. Figure 2a,b,g–j shows the morphology of Fe-based AMCs with different oxygen flows. The statistics show that the porosity of Fe-based AMCs decreases from 1.23% to 0.87% when the oxygen flow increases from 48.1 to 53.8 m^3^/h, but the porosity increases to 1.51% with the further increase in oxygen flow. In addition, when the oxygen flow increases from 48.1 to 53.8 m^3^/h, the unmelted particles appear in the Fe-based AMC and the number of unmelted particles increases with the increase of oxygen flow. Figure 2c–h shows the morphology of Fe-based AMCs with different kerosene flows. It can be found that there is a large number of pores and unmelted particles in the Fe-based AMC as the kerosene flow is 18.9 L/h. When the kerosene flow increases from 18.9 L/h to 22.7 L/h, the porosity decreases from 2.73% to 0.96% and the number of unmelted particles decreases observably. However, with the further increase in kerosene flow, the porosity of Fe-based AMC decreases slightly (from 0.96% to 0.87%) and the number of unmelted particles has no obvious change. It is worth noting that the thickness of Coating B is smaller than that of other coatings, indicating the lower deposition efficiency of the powders as the oxygen flow is 48.1 m^3^/h and the kerosene flow is 18.9 L/h.

For Fe-based AMCs, many complicated factors affect the porosity and the number of unmelted particles, but the temperature and velocity of in-flight powder particles are the most important aspects. It is reported that the in-flight molten droplets with a high velocity and opportune temperature can form good splat-spreading behaviors, which is necessary for high-quality coatings [19,20,21,22]. Oxygen can promote the full combustion of kerosene, and the combustion chamber pressure and flame temperature will increase with the increase in oxygen flow in the condition of a low oxygen/kerosene ratio, resulting in the complete fusion, higher temperature, and higher velocity of in-flight powder particles. When the kerosene has been fully burned, the increase in oxygen will not significantly increase the combustion chamber pressure but will reduce the temperature of flame and in-flight powder particles [23]. Therefore, in the HVOF process, there is an optimal proportion between oxygen and fuel where the particle temperature is moderate and the particle velocity is high. In this work, the optimum oxygen/fuel ratio is obtained. When the oxygen flow is 53.8 m^3^/h and the kerosene flow is 26.5 L/h, the Fe-based AMC (Coating D) possesses the lowest porosity and few unmelted particles. When the kerosene flow is very low but the oxygen flow is sufficient, the flame temperature and the combustion chamber pressure is lower, leading to the low velocity and unmelted state of in-flight powder particles. It is impossible for the melted or unmelted particles with low velocity to form good splat-spreading behaviors, so the porosity of Coating B is high and the deposition efficiency of the powders are low. When the kerosene flow is sufficient but the oxygen flow is very low, because kerosene is not completely burned, the flame temperature and particle velocity will be low. Therefore, the Fe-based AMC will have higher porosity and more unmelted particles, but this case does not happen in our work.

The DSC curves of the Fe-based amorphous powders and the Fe-based AMCs are shown in Figure 3. All the Fe-based AMCs and the Fe-based amorphous powders have four obvious exothermic peaks in the range of 600–924 °C. It can be found from DSC curves of Coating A, D, and E that the amorphous content of Fe-based AMCs decreases with the increase in oxygen flow. According to the DSC curves of Coating B, C, and D, the amorphous content of Fe-based AMCs increases by leaps and bounds as the kerosene flow is increased from 18.9 L/h to 22.7 L/h, but the amorphous content increases slightly with the further increase in kerosene flow. It is noteworthy that the amorphous content of Fe-based AMCs decreases with the increase in unmelted particles. The amorphous phase in Fe-based AMCs comes from the high cooling rate produced by the impact of in-flight molten droplets on the cold substrate and the amorphous phase in unmelted particles. The amorphous content of unmelted particles is lower than that of the original powders because the powder particles has been heated in flame [24]. Therefore, Coating B with a large number of unmelted particles possesses low amorphous content, and Coating A has the highest amorphous content because of the non-existence of unmelted particles.

Figure 4a,b shows the variation trends of porosity and amorphous content of Fe-based AMCs with different oxygen flow and kerosene flow. The microhardness of Fe-based AMCs with different spraying parameters are also shown in Figure 4c. It is reported that the compactness and amorphous content are the main factors affecting the properties of Fe-based AMCs. Generally, pores have a negative effect on the hardness of Fe-based AMCs, but crystallization is conducive to the improvement of hardness [25,26,27,28]. In this work, the microhardness of Fe-based AMCs increases as the porosity decreases and amorphous content increases, so the microhardness is more sensitive to the porosity, not to the amorphous content. Therefore, the microhardness of Coating D that has the lowest porosity is the highest (801 HV_0_._1_) and the microhardness of Coating B is the lowest (650 HV_0_._1_) due to its higher porosity. Figure 4d displays the bonding strength of Fe-based AMCs with different spraying parameters. Obviously, the bonding strength of Fe-based AMCs increases with the decrease in porosity. The pores in the Fe-based AMCs can cause local stress concentration as load is applied. Cracks may form preferentially near the pores, then the cracks propagate and connect with each other, resulting in the fracture of coatings. Coating D has the highest bonding strength (56.9 MPa) because of its lowest porosity. It is noteworthy that the Ni60 intermediate layer also makes a great contribution to the improvement of the bonding strength of the coating, which will be discussed later.

The potentiodynamic polarization curves of Fe-based AMCs and the LA141 magnesium alloy in 3.5 wt.% NaCl solution are shown in Figure 5a. Table 2 lists the corrosion potential (E_corr_), corrosion current density (i_corr_), pitting potential (E_pit_) and passivation current density (i_pass_) determined from the potentiodynamic polarization curves. It can be found that the corrosion potential and corrosion current density of the LA141 magnesium alloy are −1.508 V_SCE_ and 144.87 μA/cm^2^, respectively. However, the Fe-based AMCs not only exhibit higher corrosion potential and a lower corrosion current density, but also show a certain passivation ability. The corrosion current density of Fe-based AMCs decreases with the decrease in the porosity, but the corrosion potential of Fe-based AMC increases with the increase in amorphous content. The pitting potentials of Coating A, C, D, and E are similar and significantly higher than that of Coating B. Moreover, the passivation current densities of Coating B are lower than those of Coating A, C, D, and E. The pores and crystalline phases in Fe-based AMCs can destroy the compactness of passive film, and the potential difference between crystalline phases and the amorphous phase will induce pitting corrosion, so the Fe-based AMC with low porosity and high amorphous content always exhibits better corrosion resistance and pitting corrosion resistance [29,30]. Due to the low porosity and high amorphous content of Coating D, there are less defects that can destroy the compactness of passive film, such as pores and crystalline phases. The dense passivation film hinders the direct contact between the solution and the coating, so Coating D exhibits excellent corrosion resistance and passivation ability.

In order to further study the passivation ability of Fe-based AMCs, the potentiostatic measurements were also carried out at the polarization potentials of 0.5 V_SCE_, and the potentiostatic polarization curves are shown in Figure 5b. In the early stages of potentiostatic polarization, the current density of all Fe-based AMCs decreases memorably and then stabilizes gradually because of the formation of passive film and the stability of the corrosion process. The current fluctuations on the current-time curves is due to the breakdown and re-passivation of passive film on the surface of Fe-based AMCs. Nevertheless, the current changes of different coatings are different over the polarization time. When the polarization time ranges from 2000 s to 5000 s, the order of current density is as follows: Coating D < Coating C < Coating A < Coating E < Coating B. Coating A, C, and D all exhibit low porosity and high amorphous content, and their passive films are denser and have better re-passivation ability, so their polarization currents gradually stabilize at a constant value over time. Since the porosity and crystalline phases have unfavorable effects on the integrity of passive film, the polarization current density will increase with the decrease of the compactness and amorphous content of Fe-based AMCs. For Coatings E and B, the more pores and crystalline phases destroy the compactness of the passive films and the damage of passive film is faster than the repair during the polarization, so the polarization current density of Coating E and B increases over time.

The wear rates and friction coefficients of Fe-based AMCs and the LA141 magnesium alloy are shown in Figure 6. The wear rate and friction coefficient of the LA141 magnesium alloy are 3.32 × 10^−3^ mm^3^N^−1^m^−1^ and 0.66, respectively. However, the wear rates of Fe-based AMCs are less than 3 × 10^−5^ mm^3^N^−1^m^−1^ and the friction coefficients of Fe-based AMCs are lower (0.43–0.53). In addition, the lower the porosity, the higher the microhardness, and the better the wear resistance of the Fe-based AMC. Coating D has the lowest wear rate (1.91 × 10^−5^ mm^3^N^−1^m^−1^) due to its lower porosity and high hardness. In terms of the friction coefficient, the amorphous splats easily to fall off and the fresh rough friction surface are always exposed when the Fe-based AMC has higher porosity, so the friction coefficient of Coating B is about 0.53 and higher than that of other Fe-based AMCs (0.43–0.45).

## 4. Discussion

The cross-sectional TEM micrograph, SAED pattern, and element mapping of Fe-based AMC in Coating D are shown in Figure 7a,b. The Fe-based AMC is composed of splats, and obvious boundaries exist among these splats. The diffused halo ring in the SAED pattern suggests that the splats and their interface are completely amorphous. According to the EDS element mapping, the interface lacks Fe, Cr, and Mo elements, but is rich in O, indicating the existence of amorphous oxides at the interface. The oxides among the amorphous splats are inevitable due to the oxidation of molten droplets during HVOF spraying [31]. The cross-sectional TEM micrograph, SAED pattern and EDS analysis of the Ni60 intermediate layer in Coating D are also shown in Figure 7c,d. There are some diffraction spots and a diffused halo ring in the SAED pattern, which indicates that the Ni60 intermediate layer consists of both crystalline phases and amorphous phase. As shown in Figure 7c, the crystalline phases whose sizes range from 25 to 200 μm exist in the amorphous matrix. Based on the EDS analysis, the composition of the amorphous matrix is about Ni_75_._26_Cr_6_._73_Mo_4_._09_Fe_4_._85_Si_9_._04_ (at. %) but limited by the measurement device; elements B and C are not considered. The formation of amorphous phase in Ni60 intermediate layer is due to the fast cooling of molten droplets and the unique composition of Ni60 powders.

Figure 8 displays the TEM images and EDS line scanning profiles of the Fe-based AMC/Ni60 layer interface and the Ni60 layer/LA141 alloy substrate interface in Coating D. As shown in Figure 8a, no crystalline phases can be observed in the Fe-based AMC, but the Ni60 layer contains a large number of crystalline phases, which is similar to the results in the previous paragraph. According to this EDS line scanning analysis, there are continuous compositional changes at the Fe-based AMC/Ni60 layer interface and Ni60 layer/LA141 alloy substrate interface, indicating the formation of localized metallurgical bonding. The fluctuation of Ni and Cr elements in the EDS line scanning profile of the Ni60 layer confirms that the crystalline phases in the Ni60 intermediate layer are rich in Cr, and the straight lines in the EDS line scanning profile of Fe-based AMC suggests the uniform composition of amorphous splat. In Figure 8c, though the crystalline phases are not directly observed in the Ni60 intermediate layer, the obvious intensity change in the Ni and Cr elements in the EDS line scanning profile can also prove the existence of crystalline phases. In addition, there are no oxides near these interfaces due to the continuous weak intensity of the O element in the EDS line scanning profiles. During the HVOF spraying, the in-flight molten droplets with high velocity impact the substrate surface, and the formation of localized metallurgical bonding is owed to the heat carried by molten droplets and the heat generated by impact. It is essential to recognize that the high bonding strength of Coating D (56.9 MPa) is attributed to both the low porosity of Fe-based AMC and the introduction of the Ni60 intermediate layer. Because of the Fe/Mg immiscible systems and the huge performance difference between Fe-based AMC and magnesium alloy, the bonding strength of the Fe-based AMC/magnesium alloy substrate interface is always low [16]. Similar to the phenomenon of the “Cask Effect”, the failure of the coating depends on the weakest area rather than the strongest area. The addition of a Ni60 intermediate layer can avoid the weak interface between the Fe-based AMC and the LA141 alloy, and the cohesive strength of Fe-based AMC is lower than the bonding strength of these interfaces and the cohesive strength of the Ni60 intermediate layer. Therefore, the bonding strength of the coating depends on the cohesive strength of Fe-based AMC in this work, and the dense Fe-based AMC usually has high cohesive strength. The effect of a Ni60 intermediate layer on the properties of Fe-based AMC has been investigated and discussed in another paper (under review) written by the current author.

The electrochemical impedance behavior of Coating D and the LA141 magnesium alloy were also measured to evaluate the effect of Fe-based AMC on the corrosion resistance of the LA141 magnesium alloy. As shown in Figure 9a,b, the Nyquist plot of Coating D is constitutive of two capacitive loops, but the Nyquist plot of the LA141 magnesium alloy consists of one capacitive loop and one inductive loop. The Nyquist plots of Coating D and the LA141 magnesium alloy are quite different, indicating their different corrosion mechanisms. The schematic diagram of the corrosion process and fitted equivalent circuits of Coating D and the LA141 magnesium alloy are shown in Figure 9c,d. The equivalent circuit of Coating D consists of the electrolyte solution resistance (R_s_), the sum of the coating resistance and the solution resistance at the unavoidable pores or cracks of coating (R_c_), the CPE of coating (CPE_c_), the CPE related to the electric double layer of the electrolyte/coating interface in the pores or cracks (CPE_dl_), and the charge transfer resistance across the electric double layer of the electrolyte/coating interface (R_t_) [32]. In the equivalent circuit of the LA141 magnesium alloy, R_L_ and L are related to the resistance of a Mg+ reaction and the inductance of a Mg+ reaction on the breaking area of the discontinuous protective film, respectively. Furthermore, CPE_dl_ represents the capacitor of the electric double layer at the solution/LA141 alloy interface, and R_t_ refers to the charge transfer resistance from the alloy surface to the solution [33]. The EIS fitted results for Coating D and the bare LA141 magnesium alloy are listed in Table 3. According to the impedance modulus, the ${\left|Z\right|}_{f=0.01\;\mathrm{Hz}}$ of Coating D is obviously larger than that of the LA141 magnesium alloy substrate, which suggests that the Fe-based AMC exhibits better corrosion resistance and has the ability to provide excellent corrosion protection for the LA141 alloy substrate.

In order to study the mechanism of the corrosion of Fe-based AMC and the LA141 magnesium alloy, Coating D partially covered by epoxy resin was corroded in 3.5 wt.% NaCl solution at 1.3 V_SCE_ for 30 min, and the LA141 magnesium alloy was immersed in 3.5 wt.% NaCl solution at the open circuit potential for 30 min. The corroded and uncorroded surface morphologies of Coating D and LA141 magnesium alloy are shown in Figure 10. It can be found that the protected area of Fe-based AMC is flat, but the eroded area is bumpy, like ravines. There is a Cr-depleted zone at the intersplat region where pitting takes place preferentially because of the oxidation of molten droplets [34]. Therefore, the concave region on the corroded surface is actually the intersplat region, and the intersplat regions can act as corrosion evolutionary paths and facilitate the corrosion medium entering into the protection area and the interior of the coating. The surface of LA141 magnesium alloy without corrosion is very smooth with only some casting defects. After immersion in 3.5wt.% NaCl solution for 30 min, there are lots of holes on the surface of the LA141 magnesium alloy, indicating the serious pitting corrosion. Therefore, for Fe-based AMC, the intersplat regions are corroded preferentially, and the corrosive solution penetrates into the coating through these intersplat regions, while the LA141 magnesium alloy is corroded mainly in the form of pitting corrosion.

The morphology and elemental distribution of the worn surface of Coating D are shown in Figure 11a–c. Many black sheets are on the uneven worn surface of Fe-based AMC, and some small pieces that are about to fall off exist at the edge of them. According to the EDS mapping, the areas of black sheets lack the elements of Fe, Cr and Mo, but are rich in the O element, so the black sheets can be confirmed as oxides. In addition to the oxides, there are also some rugged areas caused by the exfoliated amorphous splats, and some smooth areas aroused by the constant friction on the worn surface. Figure 11d–f exhibits the morphology of the wear debris of Coating D and the element concentration of the specified zone. There are not only fine crumbs, but also large lamellae in the wear debris. The shape and size of large lamellae are indefinite, but cracks and small pieces about to fall off exist at their edges. Moreover, the large lamellae can be divided into two types, one with furrows, and the other with a smooth surface. According to the element concentration, the oxygen content of the fine crumbs is the highest, followed by the large lamellae with furrows, and the oxygen content of large lamellae with smooth surface is almost zero. Therefore, it can be inferred that the surface with furrows and the surface without furrows are the front and back of the exfoliated amorphous splats, respectively. In the wear process, the front of the amorphous splats constantly rubs against the Si_3_N_4_ ball, resulting in oxidation at a high temperature [35]. The back of the amorphous splats does not have contact with the friction pair, so the oxygen content on the back of the amorphous spats is much lower than that on the front. Furthermore, the shapes of fine crumbs in the wear debris are very similar to those of the small pieces at the edge of the oxides, which indicates that fine crumbs may be formed by the smash of oxides during wear. As shown in Figure 11g, some cracks exist among the amorphous splats and some amorphous splats are fractured. Because the intersplat regions are relatively weak, the cracks preferentially occur at them and propagate along them under the effect of shear force, leading to the falling off of amorphous splats. Additionally, the wear with the friction pair will cause the deformation and fracture of the outermost amorphous splats. The morphology of the worn surfaces of the LA141 magnesium alloy is shown in Figure 11h,i. The wear scar width of LA141 magnesium alloy is much larger than that of Fe-based AMC, and there are many furrows and some scallops caused by the exfoliated lamellar on the worn surface. According to the EDS line scanning, the worn surface is rich in the O element. The oxidation of the worn surface is due to the high temperature produced by the friction. As a result, the mechanism on the wear of Fe-based AMC is mainly fatigue wear and oxidative wear, and the mechanism on the wear of the LA141 magnesium alloy is mainly abrasive wear, oxidative wear and adhesive wear. The better corrosion and wear resistance of Fe-based AMCs in this study can provide a new and effective protection for the LA141 magnesium alloys.

## 5. Conclusions

The HVOF-sprayed Fe-based AMCs with a Ni60 intermediate layer have been prepared on the substrate of an LA141 magnesium alloy successfully. To obtain the high performance of Fe-based AMC, the impact of oxygen flow and kerosene flow on the microstructure and performance of the Fe-based AMC are investigated. The following conclusions can be drawn:(1)The porosity of Fe-based AMCs decreases with the increase in kerosene flow, and there is an optimal oxygen/kerosene ratio, but the amorphous content of Fe-based AMCs increases with the decrease in the oxygen/kerosene ratio. Moreover, the increase in porosity and the decrease in amorphous content have negative effects on the hardness, bonding strengths, corrosion resistance and wear resistance of Fe-based AMC, but the properties are more sensitive to the porosity.(2)When the oxygen flow is 53.8 m^3^/h and the kerosene flow is 26.5 L/h, the Fe-based AMC exhibits the optimum performance, such as low porosity (0.87%), high amorphous content (90%), high microhardness (801 HV), high bonding strength (56.9 MPa), and outstanding corrosion and wear resistance.(3)The Fe-based AMC is composed of amorphous splats and amorphous oxides, but the Ni60 intermediate layer consists of both amorphous and crystalline phases. Localized metallurgical bonding can be formed at the Fe-based AMC/Ni60 layer interface and the Ni60 layer/LA141 alloy substrate interface. The high bonding strength of the coating is attributed to the low porosity of Fe-based AMC and the introduction of the Ni60 intermediate layer.(4)In the corrosion process of Fe-based AMC, the intersplat regions that lack Cr elements are corroded preferentially and act as corrosion evolutionary paths, leading to the invasion of corrosive solution. The mechanism on the wear of Fe-based AMC is mainly fatigue wear and oxidative wear, and the mechanism on the wear of LA141 magnesium alloy is mainly abrasive wear, oxidative wear, and adhesive wear.

## Figures and Tables

**Figure 1 materials-14-04786-f001:**
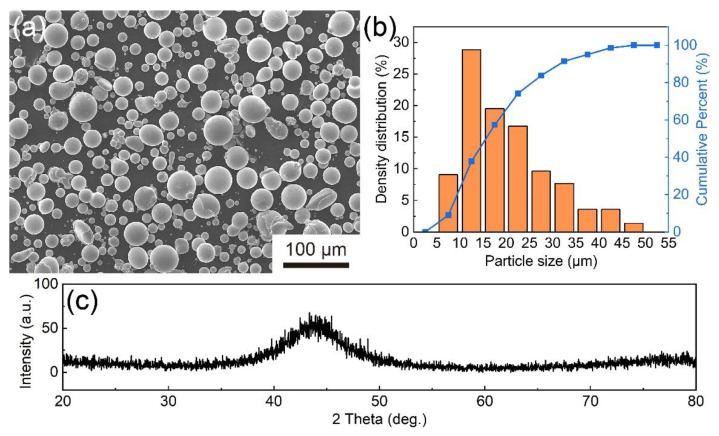
SEM image (**a**), size distribution (**b**) and XRD pattern (**c**) of Fe-based amorphous powders.

**Figure 2 materials-14-04786-f002:**
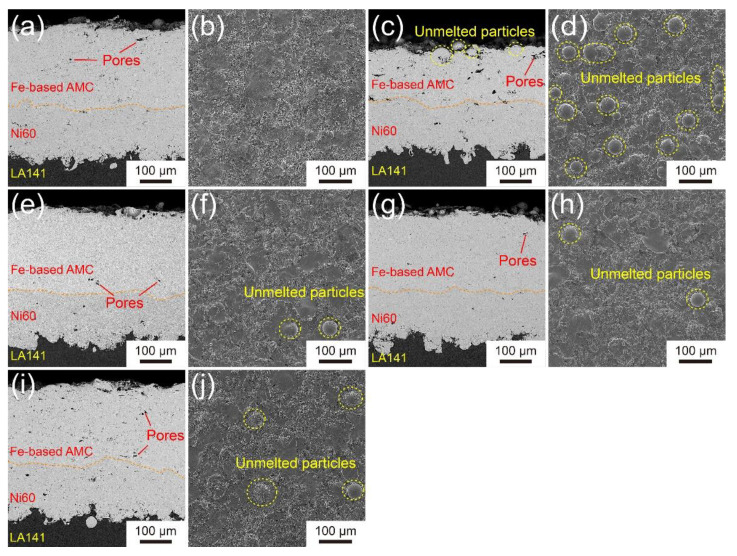
Cross-sectional images and surface images of the Fe-based AMCs on LA141 magnesium alloy: (**a**,**b**) Coating A, (**c**,**d**) Coating B, (**e**,**f**) Coating C, (**g**,**h**) Coating D and (**i**,**j**) Coating E.

**Figure 3 materials-14-04786-f003:**
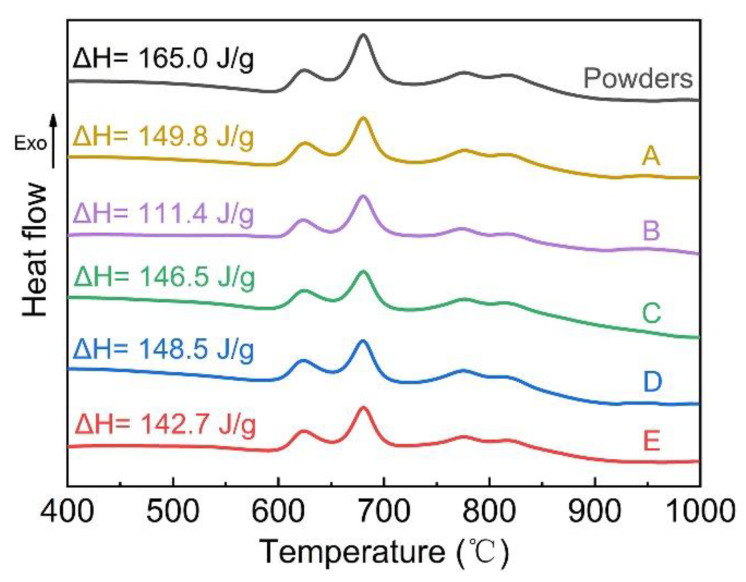
DSC curves of Fe-based amorphous powders and Fe-based AMCs.

**Figure 4 materials-14-04786-f004:**
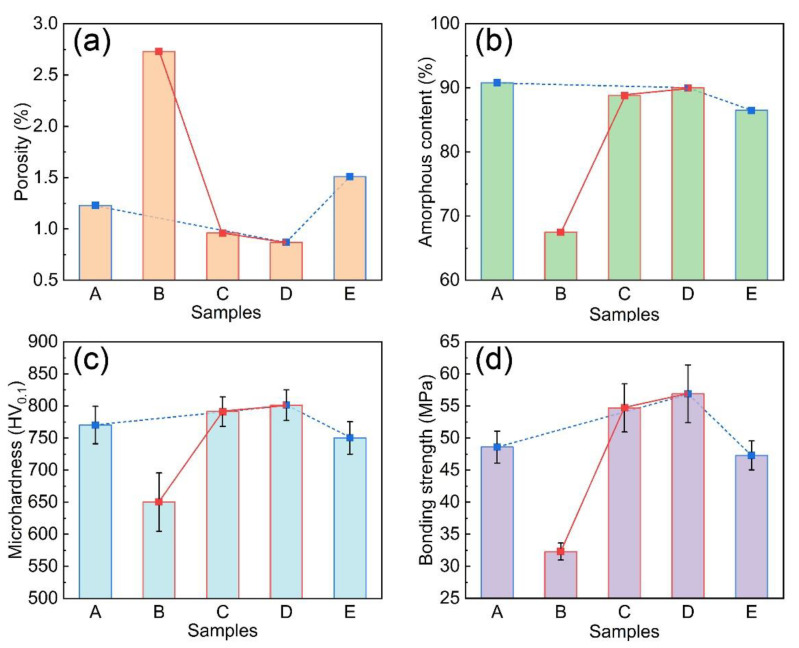
Porosity (**a**), amorphous content (**b**), microhardness (**c**), and bonding strength (**d**) of Fe-based AMCs on LA141 magnesium alloy.

**Figure 5 materials-14-04786-f005:**
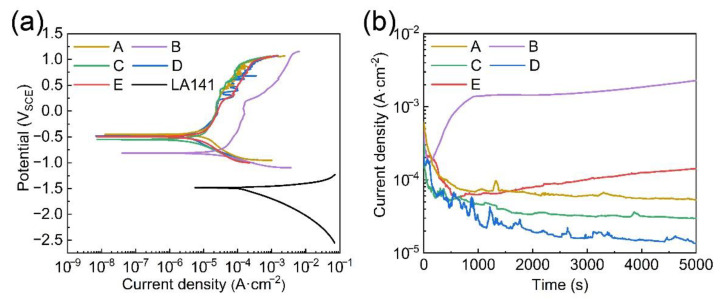
(**a**) Potentiodynamic polarization curves of Fe-based AMCs and LA141 magnesium alloy; (**b**) potentiostatic polarization curves of Fe-based AMCs.

**Figure 6 materials-14-04786-f006:**
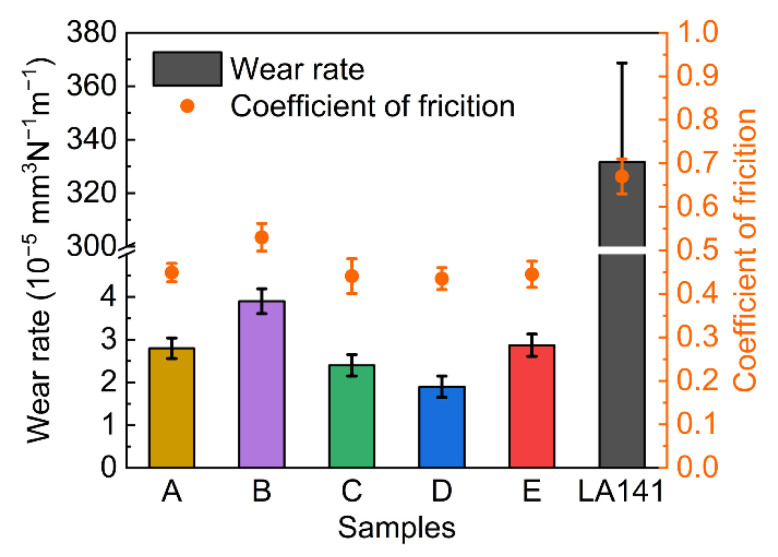
Wear rate and friction coefficient of Fe-based AMCs and LA141 magnesium alloy.

**Figure 7 materials-14-04786-f007:**
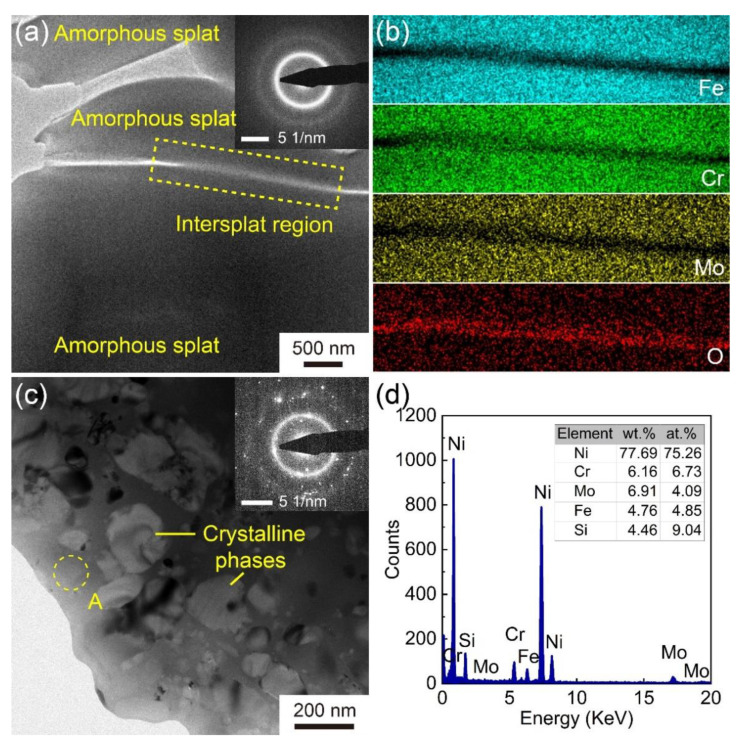
(**a**) Cross-sectional TEM image of Fe-based AMC in Coating D; inset in (**a**) is the SAED pattern taken from the intersplat region; (**b**) EDS element mapping of the intersplat region; (**c**) cross-sectional TEM images of Ni60 intermediate layer in Coating D; inset in (**c**) is the SAED pattern taken from the Ni60 splats; (**d**) EDS spectra of point A in (**c**).

**Figure 8 materials-14-04786-f008:**
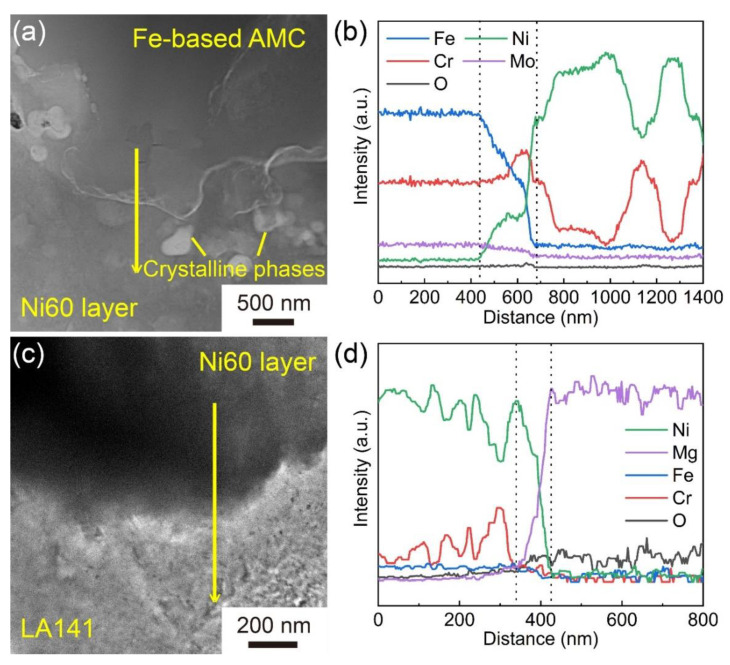
TEM image (**a**) and corresponding EDS line scanning profiles (**b**) of the Fe-based AMC/Ni60 layer interface in Coating D. TEM image (**c**) and corresponding EDS line scanning profiles (**d**) of the Ni60 layer/LA141 alloy substrate interface in Coating D.

**Figure 9 materials-14-04786-f009:**
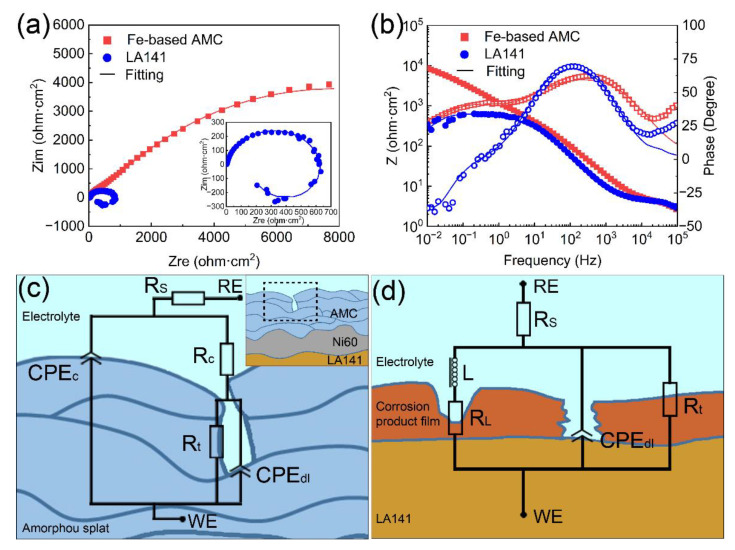
Electrochemical impedance behavior of Coating D and the LA141 magnesium alloy substrate: (**a**) Nyquist plots, (**b**) Bode impedance magnitude plots and Bode phase angle plots. (**c**) Fitted equivalent circuit model and schematic diagram of corrosion process of Coating D. (**d**) Fitted equivalent circuit model and schematic diagram of corrosion process of LA141 magnesium alloy.

**Figure 10 materials-14-04786-f010:**
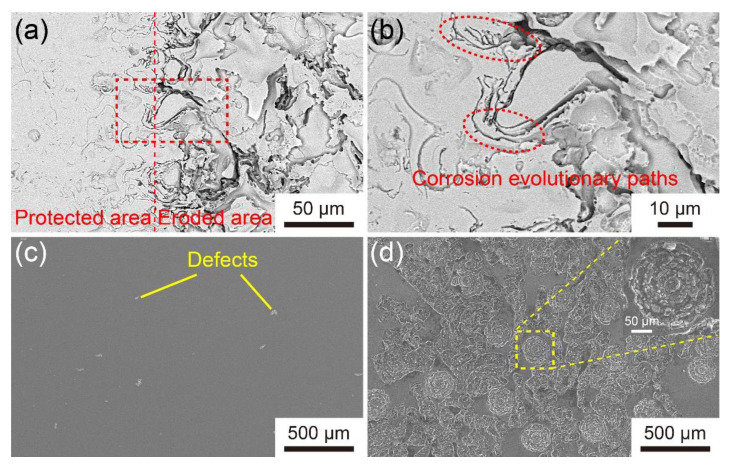
SEM image with low (**a**) and high (**b**) magnification of the protected area and eroded area on Coating D after potentiostatic polarization at 1.3 V_SCE_ for 30 min in 3.5 wt.% NaCl solution; (**c**) SEM image of the uncorroded surface of LA141 magnesium alloy, (**d**) SEM image of the corroded surface of LA141 magnesium alloy after immersion in 3.5 wt.% NaCl solution at the open circuit potential for 30 min. Inset in (**d**) is higher magnification SEM micrograph of pitting.

**Figure 11 materials-14-04786-f011:**
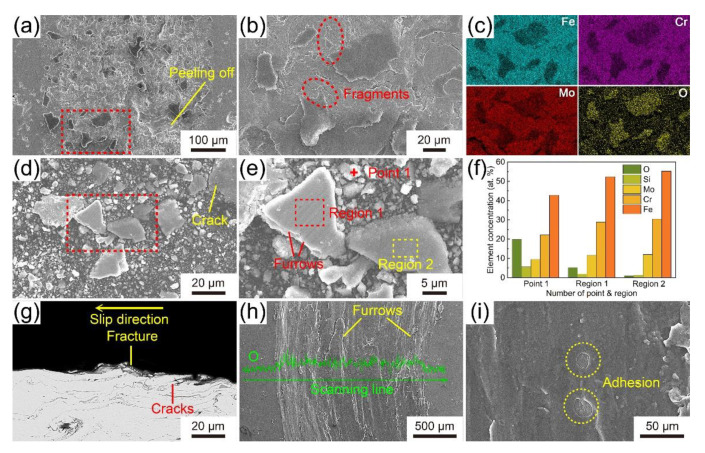
SEM images with low (**a**) and high (**b**) magnification of the worn surfaces of Coating D after abrasion, (**c**) EDS mapping of the zone marked in (**a**); SEM images with low (**d**) and high (**e**) magnification of the wear debris of Coating D; (**f**) element concentration of Point 1, Region 1 and Region 2 in (**e**); (**g**) SEM image of the cross section of Coating D after abrasion; SEM images with low (**h**) and high (**i**) magnification of worn surfaces of LA141 magnesium alloy; insets of (**h**) are the EDS line-scanning pattern of O element along the marked direction.

**Table 1 materials-14-04786-t001:** Detailed spraying parameters of Fe-based AMCs.

Coatings	A	B	C	D	E
Oxygen flow (m^3^/h)	48.1	53.8	53.8	53.8	59.5
Kerosene flow (L/h)	26.5	18.9	22.7	26.5	26.5
Spray distance (mm)	350	350	350	350	350
Powder feed rate (g/min)	80	80	80	80	80
Scanning velocity (mm/s)	300	300	300	300	300

**Table 2 materials-14-04786-t002:** Corrosion data determined from the potentiodynamic polarization curves.

Samples	E_corr_ (V_SCE_)	I_corr_ (μA/cm^2^)	E_pit_ (V_SCE_)	I_pass_ (μA/cm^2^)
Coating A	−0.447	9.95	0.931	25.97
Coating B	−0.839	33.49	0.204	115.2
Coating C	−0.627	7.35	0.911	24.89
Coating D	−0.528	4.23	0.945	25.59
Coating E	−0.556	9.36	0.936	37.13
LA141 alloy	−1.508	144.87	-	-

**Table 3 materials-14-04786-t003:** EIS fitted results for Coating D and the bare LA141 magnesium alloy.

Sample	LA141 Magnesium Alloy	Coating D
R_s_ (Ω·cm^2^)	3.61	2.90
R_t_ (Ω·cm^2^)	643	14,300
CPE_dl_ (F·cm^−2^·s^n−1^)	8.37 × 10^−5^	2.95 × 10^−4^
CPE_dl_-n	0.820	0.578
R_c_ (Ω·cm^2^)	/	9946
CPE_c_ (F·cm^−2^·s^n−1^)	/	9.59 × 10^−5^
CPE_c_-n	/	0.721
R_L_ (Ω·cm^2^)	22.88	/
L (H·cm^2^)	4943	/
|Z|_f = 0.01 Hz_ (Ω·cm^2^)	273	8691

## Data Availability

Data are contained within the article and can be requested from the corresponding author.

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
