# Peer review of "High-Performance HVOF-Sprayed Fe-Based Amorphous Coating on LA141 Magnesium Alloy via Optimizing Oxygen Flow and Kerosene Flow"

_materials, 2021, doi:10.3390/ma14174786_

Round 1

Reviewer 1 Report

are these amorphous layer, I say these are nanocrystalline, fig. 1c show XRD-diagramm,, position near 110 peak, FWHD app. 2Theta=7° this means in Scherrer equation 2 nm. Intensity scale isn t shown, measured with Cu K-alpha radiation produces high fluorescence amount.

The article have many not correct tasks/descriptions:

  • scfh and gph are not SI units - change this in SI units
  • what type of detector is used in XRD
  • hardness measurements load (100g) is not correct, now it must Newton
  • page 4 and --> how to estimate the amorphous content?
  • I not see, that diffraction SAED in fig 7c is an amorphous, spots of big grains are to seen, the can be arranged on a ring --> crystallinity occurs.
  • there is no remark, how often the authors repeat the experiments
  • my Opinium: they investigate only the five different conditions. 
  • only many experiments the done with sample D
  • so the results on page. 10 - 14 are not comparable to other experimental conditions. 

Author Response

Dear Reviewer:

Thank you for your comments concerning our manuscript entitled “High-performance HVOF-sprayed Fe-based amorphous coating on LA141 magnesium alloy via optimizing oxygen flow and kerosene flow” (ID: materials-1303058). Those comments are all valuable and very helpful for revising and improving our paper, as well as the important guiding significance to our researches. We have studied comments carefully and have made correction which we hope to meet your journal with an approval. The main corrections in the paper and the responds to the reviewer’s comments can be found in the attachment.

Reviewer 2 Report

The paper ”High-performance HVOF-sprayed Fe-based amorphous coating on LA141 magnesium alloy via optimizing oxygen flow and kerosene flow” is suitable for publication in Materials Journal. It has, in my opinion, all the experimental data for complete microstructural analysis: SEM, TEM, XRD, EDS analysis, mapping and electrochemical evaluation. I agree with the publication. 

Author Response

Dear Reviewer:

Thank you for your comments concerning our manuscript entitled “High-performance HVOF-sprayed Fe-based amorphous coating on LA141 magnesium alloy via optimizing oxygen flow and kerosene flow” (ID: materials-1303058). We are honored that this paper can be recognized by you, and we will do more research in the field of coating on light alloy substrate.

Reviewer 3 Report

The paper deals with the coating of Fe-based amorphous layer on LA141 Mg alloy through HVOF process and is considered authors’ own work. The reviewer believes that the paper is possible to include the journal after minor modifications. The followings are the comment to the authors.

  1. The use of SI unit is recommended all through the manuscript.
  2. The use of superscript and subscript should be modified appropriately. Check carefully all through the manuscript.
  3. Line 56 and Line 59; "Tian et al" and "Zhang et al", period needed for both end.
  4. “2. Materials and Methods”; It is requested to mention how the first Ni60 layer was coated followed by the coating of Fe-based amorphous layer in details. Nothing details are demonstrated.
  5. Line 116 and later; Time or length of wear test should be written appropriately.
  6. Figure 4; How the authors measured the porosity which should be included in the manuscript. Also, the method to measure the bonding strength should be written in the manuscript.
  7. Line 292; “…elements B and C…” seems not clear for the reviewer.
  8. Line 329-330; “…has been investigated and discussed in details elsewhere written by the current authors.”; The paper should be cited in the references if the paper has been published. Make it clear the status, please.

Author Response

(The authors gave the same response as above.)

Round 2

Reviewer 1 Report

The authors write:

The Fe-based AMC is composed of amorphous splats and amorphous oxides, but the Ni60 intermediate layer consists of both amorphous and crystalline phases. Localized metallurgical bonding can be formed at Fe-based AMC/Ni60 layer interface and Ni60 layer/LA141 alloy substrate interface.

I see not the crystallinity in a XRD diagram of Ni60!

table 3 must be better designed, 

Author Response

(The authors gave the same response as above.)
